# Force-Driven Model for Automated Clear Aligner Staging Design Based on Stepwise Tooth Displacement and Rotation in 3D Space

**DOI:** 10.3390/bioengineering12020111

**Published:** 2025-01-25

**Authors:** Sensen Yang, Yumin Cheng

**Affiliations:** 1Shanghai Key Laboratory of Mechanics in Energy Engineering, Shanghai Institute of Applied Mathematics and Mechanics, School of Mechanics and Engineering Science, Shanghai University, Shanghai 200072, China; yangsensen@smartee.cn; 2Smartee Biomechanics Research Laboratory, Shanghai Smartee Denti-Technology, Shanghai 201210, China

**Keywords:** clear aligner, force-driven model, automated staging, treatment planning, stepwise movement control, tooth displacement, tooth rotation, extraction cases, anchorage control, orthodontic software

## Abstract

This study introduced a novel force-driven automated staging design method for clear aligners, aimed at enhancing treatment planning efficiency and outcomes. The method simplified the alignment process into a force-driven mechanics model that calculates forces and moments exerted on teeth while adhering to Newton’s third law, determining their displacement and rotation at each position. An optimal path was generated by iteratively moving teeth from their initial to target positions and subsequently divided into stages based on a predefined step size. The algorithm was implemented in C++ and incorporated into the WebGL-based SmarteeCheck3.0 software for visualization. In a maxillary extraction case, the automated staging method (0.25 mm step size) generated 51 stages in merely 5 s, while manual staging (>0.25 mm step size) necessitated 30 min to achieve 55 stages. In a molar distalization case, the automated method demonstrated similar efficiency advantages, generating 30 stages for the maxilla and 34 for the mandible, compared to 41 stages each in manual staging. The automated staging approach yielded shorter and more precise tooth movement paths that adhered to aligner biomechanics and physical principles, surpassing the limitations of manual staging. For cases requiring entire arch displacement, the method incorporated sequential movements with anchorage control to maintain force equilibrium. This innovative method substantially improved design efficiency and accuracy, ultimately elevating the efficacy of clear aligner therapy, although further biomechanical analyses and experimental validations are needed to refine the model parameters.

## 1. Introduction

Aligner orthodontics has become the most popular orthodontic treatment, but its staging design remains a challenge [1,2]. Staging is the step-by-step progression of tooth movement from the initial position to the target position, and it primarily relies on manual efforts. As most clinicians are not proficient in applying the software for staging design, they are limited to designing the target position and providing general guidance. Consequently, they must rely heavily on technicians from aligner manufacturers for detailed staging design. But technicians are not directly involved in patient communication and must work with second-hand information from clinicians, further reducing efficiency and the accurate representation of clinical situations. Each technician typically spends a minimum of 30 min per case on staging, resulting in low efficiency and high personnel costs, which can ultimately impact the affordability and accessibility of clear aligner treatment.

Staging also plays a crucial role in the efficacy of clear aligner treatment, which is measured by the ratio of actual tooth movement to the designed tooth movement. On average, this efficacy reaches only about 50% [3,4,5,6]. This relatively low efficacy highlights why accurate staging design is particularly important for clear aligners because, unlike fixed appliances where clinicians can make real-time adjustments based on treatment progress [7], clear aligner therapy requires comprehensive pre-planning as the entire treatment sequence is manufactured in advance. For the same case, different technicians may propose varying staging designs. These differences primarily stem from the technicians’ knowledge and experience and inevitably lead to variations in efficacy. Furthermore, technicians face challenges in accurately calculating and analyzing tooth force systems mentally within a short time, limiting them to qualitative judgments and hindering the development of precise tooth movement plans. To address these issues and enhance the effectiveness, consistency, and accuracy of clear aligner treatment, a transition from manual design to advanced algorithms for automated staging is necessary.

Currently, only a few clear aligner manufacturers have developed automated staging design capabilities. Invisalign claims to have automated staging functionality and has proposed force-driven methods, its theories and algorithms are opaque, with internal computational mechanisms unknown to others, and its efficacy remains suboptimal [8,9,10,11,12]. Although many researchers have adopted the term “force-driven” as their core methodology since using force analysis to investigate orthodontic problems is a first-principle approach, their actual methods vary considerably. Existing studies on automated tooth arrangement focus on target position arrangement rather than automated staging design, revealing a research gap. Target position arrangement is based on dental aesthetics [13,14], Andrews’ six keys to normal occlusion [15], standard arch forms [16,17], and geometry-related deep learning techniques [18,19,20,21,22], while staging design research mainly employs robotics-related path planning methods [23]. However, all these methods implicitly apply external force to the teeth, which does not conform to the fundamental principles of aligner biomechanics. In aligner biomechanics, the force exerted by an aligner on a tooth must come from the opposing reactive force of other anchorage teeth, and all these forces have an equivalent effect on tooth movement [24,25,26,27]. If this crucial aspect is not given due attention, it can lead to severe anchorage loss issues. Many manual and automated methods have failed to take this fundamental physical law into consideration.

This study proposes a simplified force-driven model that adheres to the fundamental principles of aligner biomechanics. The model simulates the aligner–teeth force system by calculating the forces and moments acting on each tooth at its current position and iteratively determines the incremental displacement and rotation of each tooth, mapping out the movement trajectory from initial to target positions. The movement trajectory is then divided into stages based on the step size. By closely adhering to the physical law, this method can effectively improve the accuracy of staging design.

## 2. Materials and Methods

The alignment process of aligners can be simplified into a mechanics model that calculates the forces and moments acting on teeth, while adhering to Newton’s third law to maintain force equilibrium. Assuming the target positions have been determined (Figure 1a), the current position represents any intermediate position during the alignment process (Figure 1b). Each tooth is assigned a local coordinate system that moves and rotates along with the tooth, used to characterize its position and orientation. By controlling the translation and rotation of these local coordinate systems, the complete tooth movement path can be determined. For simplicity and clarity, teeth are represented by planar diagrams, although they are three dimensional (3D) in reality. Teeth are sequentially numbered (non-FDI notation) to facilitate formula programming.

The displacement design of aligners generates a driving force that propels teeth towards their target positions. However, this force does not manifest or dissipate out of thin air. The driving force responsible for moving the target tooth inevitably originates from the adjacent teeth, and in accordance with Newton’s third law, the action and reaction forces are equal in magnitude but opposite in direction. In Figure 1b, tooth 5 deviates from its target position. Consider teeth 4, 5, and 6 as an illustrative example (Figure 2a) to calculate the force acting on tooth 5 during the alignment process. During the process, the aligner experiences a reverse correction moment $Kt\times \theta_{45}$, where $\theta_{45}$ represents the angle between the current and target positions of the line connecting teeth 4 and 5 in 3D space, the moment direction is obtained by calculating the cross product of the current and target positions of the line connecting teeth 4 and 5, and Kt represents the angular stiffness. This correction moment is expressed as equal and opposite forces $Fq_{54}$ and $Fq_{45}$ acting on the adjacent teeth (Figure 2b) and they all have components in the x, y, and z directions. Similarly, for teeth 5 and 6, forces $Fq_{56}$ and $Fq_{65}$ are obtained.(1)$$Fq_{45}=-Fq_{54}=\frac{Kt\times \theta_{45}}{l_{54}}$$(2)$$Fq_{56}=-Fq_{65}=\frac{Kt\times \theta_{56}}{l_{56}}$$

The above calculations can determine the relative orientations of the teeth in 3D space, but their relative distances have not yet been determined. Next, we need to determine the distances between them. When the distance between adjacent teeth is greater than the distance between adjacent teeth in the target position, the adjacent teeth experience equal magnitude and opposite direction tensile forces $Ft_{54}$ and $Ft_{45}$. For teeth 5 and 6, these forces are $Ft_{56}$ and $Ft_{65}$ (Figure 2c). When the distance between adjacent teeth is smaller than the distance between adjacent teeth in the target position or when teeth collision occurs, the adjacent teeth experience equal magnitude and opposite direction collision forces. For teeth 4 and 5, these forces are $Fc_{54}$ and $Fc_{45}$, and for teeth 5 and 6, these forces are $Fc_{56}$ and $Fc_{65}$ (Figure 2d). Finally, the resultant force acting on tooth 5 (Figure 2e) can be obtained as(3)$$F_{5}=Fq_{54}+Fq_{56}+Ft_{54}+Ft_{56}+Fc_{54}+Fc_{56}$$

By performing the calculation for each tooth sequentially, the forces acting on each tooth are obtained as(4)$$F_{i}=Fq_{i,i-1}+Fq_{i,i+1}+Ft_{i,i-1}+Ft_{i,i+1}+Fc_{i,i-1}+Fc_{i,i+1}\;(i=1,2,\cdots ,14)$$

Assuming the translational stiffness of each tooth is $Kzt_{i}$, $Kzt_{i}$ is related to the area and morphology of the tooth root, and in preliminary calculations, it can be approximated using the average root surface area derived from statistical data. The movement vectors of all teeth in their current positions are(5)$$d=\left(\frac{F_{1}}{Kzt_{1}},\frac{F_{2}}{Kzt_{2}},\cdots ,\frac{F_{14}}{Kzt_{14}}\right)$$

The above calculations determine the relative positions of the tooth coordinate system’s origin, but not the orientation angles of each tooth coordinate system. Next, the moments acting on the teeth and their rotation angles are calculated, assuming that the current angle between teeth 4 and 5 is $\phi_{R45}$, and the target angle is $\phi_{T45}$ (Figure 3a). For comparing angles in 3D space, it can be achieved through projections onto two orthogonal planes. Once the orientations of two axes are determined, the orientation of the third axis is naturally determined. During the alignment process, the two teeth experience a pair of equal magnitude and opposite direction moments $M_{45}$ and $M_{54}$, which bring their angle closer to the target angle. Similarly, teeth 5 and 6 experience moments $M_{56}$ and $M_{65}$, respectively (Figure 3b). Assuming the moments are proportional to the angular difference, with the corresponding angular stiffness $K_{r}$, initially, they can be set as equal, and more precise values can be obtained through iterative testing and finite element analysis. The corresponding moments are as follows:(6)$$M_{54}=-M_{45}=\frac{K_{r}\times (\phi_{R45}-\phi_{T45})}{2}$$(7)$$M_{56}=-M_{65}=\frac{K_{r}\times (\phi_{R56}-\phi_{T56})}{2}$$

The resultant moment acting on tooth 5 is as follows:(8)$$M_{5}=M_{54}+M_{56}$$

By performing the calculation for each tooth sequentially, the moments acting on each tooth are obtained as(9)$$M_{i}=M_{i,i-1}+M_{i,i+1}\;(i=1,2,\cdots ,14)$$

Assuming the rotational stiffness of each tooth is $Kzr_{i}$, which corresponds to the moment per unit angle of tooth rotation, the rotation angles of all teeth are as follows:(10)$$\varphi =\left(\frac{M_{1}}{Kzr_{1}},\frac{M_{2}}{Kzr_{2}},\cdots ,\frac{M_{14}}{Kzr_{14}}\right)$$

Furthermore, there are many methods to simplify the control of tooth rotation. For example, the teeth can be directly controlled to rotate towards the target position, and the reaction moments can be evenly distributed to the anterior and posterior teeth.

As the forces and moments acting on the teeth vary constantly depending on their position, the tooth movement must be progressively updated through small, iterative displacements. The minimum displacement for each iteration can be set to 0.01 mm, and the minimum rotation angle can be set to 0.04°. To obtain the iterative displacement Δd and rotation angle Δφ for each iteration, the displacements and angles of all teeth are normalized proportionally and then multiplied by the corresponding minimum displacement and angle.(11)Δd=0.01 mm×Normalize{d}(12)$$\Delta \varphi =0.04^{o}\times Normalize\{\varphi \}$$

Tooth displacements and rotations occur in 3D space (Figure 4). To describe the angular rotation between two rigid bodies in 3D space, the axis-angle representation is utilized. A 4 × 4 matrix is employed to represent all translations and rotations simultaneously [28]. Throughout the simulation process, each tooth is assigned a 4 × 4 transformation matrix $T_{i}$ in every iteration to accurately capture and represent its movement in 3D space.(13)$$T_{i}=\left[\begin{array}{cccc} f_{ix}f_{ix}vc\Delta \varphi_{i}+c\Delta \varphi_{i} & f_{iy}f_{ix}vc\Delta \varphi_{i}-f_{iz}s\Delta \varphi_{i} & f_{iz}f_{ix}vc\Delta \varphi_{i}+f_{iy}s\Delta \varphi_{i} & \Delta d_{ix}\\ f_{ix}f_{iy}vc\Delta \varphi_{i}+f_{iz}s\Delta \varphi_{i} & f_{iy}f_{iy}vc\Delta \varphi_{i}+c\Delta \varphi_{i} & f_{iz}f_{iy}vc\Delta \varphi_{i}-f_{iy}s\Delta \varphi_{i} & \Delta d_{iy}\\ f_{ix}f_{iz}vc\Delta \varphi_{i}-f_{iz}s\Delta \varphi_{i} & f_{iy}f_{iz}vc\Delta \varphi_{i}+f_{ix}s\Delta \varphi_{i} & f_{iz}f_{iz}vc\Delta \varphi_{i}+c\Delta \varphi_{i} & \Delta d_{iz}\\ 0 & 0 & 0 & 1 \end{array}\right]$$
where $f_{i}={\left(f_{ix},f_{iy},f_{iz}\right)}^{T}$ is the rotation axis unit vector, $vc\Delta \varphi_{i}=1-\cos\Delta \varphi_{i}$, $c\Delta \varphi_{i}=\cos\Delta \varphi_{i}$ and $s\Delta \varphi_{i}=\sin\Delta \varphi_{i}$.

To calculate the forces and moments acting on the teeth, an iterative process is employed, updating the tooth positions with each iteration until the desired target position is achieved. Typically, completing the teeth’s movement trajectory requires 500 to 1000 iterations to ensure a smooth and accurate progression. The stepwise advancement is determined by tracking the maximum displacement of each tooth and recording a new step whenever a cumulative displacement of 0.25 mm is reached. This approach allows for precise monitoring and control of the teeth’s movement. At each step, the displacement and angular changes of the teeth are derived from the corresponding transformation matrices, which provide a comprehensive and detailed record of the teeth’s movement throughout the entire trajectory. This record enables orthodontists and researchers to analyze and evaluate the efficiency and effectiveness of the treatment plan, making adjustments as necessary to optimize the outcome for each individual patient.

## 3. Results

We developed an implementation of the algorithm in C++ and incorporated it into the WebGL-based SmarteeCheck3.0 software, enabling the display of corresponding animations on web pages. To illustrate the effectiveness of the approach, we conducted a comparison between the force-driven automated staging and doctor-guided manual staging using an extraction case as an example (Figure 5). The automated staging, which employs a precise step size of 0.25 mm for the maximum displacement of the crown (accounting for both translational and rotational displacement), yielded 51 stages for the upper arch and 39 stages for the lower arch (Figure 5a), with simultaneous movement of all teeth. In contrast, the manual staging design exhibited unequal step sizes, with the maximum step size exceeding 0.25 mm, resulting in 55 stages for the upper arch and 50 stages for the lower arch (Figure 5b). To facilitate a more comprehensive understanding of the differences between the two approaches, we have provided URLs [29,30] (these URLs are only accessible from Chinese IP addresses) in the figures, allowing readers to access and view the staging animations for each method.

Figure 6 and Figure 7 showcase the different stages of automated and manual staging, respectively, highlighting the distinct approaches employed in each method. In manual staging, tooth 13 (canine) is initially moved along a large round-trip path to create adequate space before reaching its final position. Only after sufficient space is available does tooth 12 (lateral incisor) begin its movement. Conversely, the force-driven automated staging moves teeth 13 and 12 simultaneously; as tooth 13 creates space, tooth 12 concurrently occupies the newly created space. This concurrent movement approach eliminates the waiting time required for space creation in manual staging, ultimately resulting in shorter travel distances for the teeth and a more efficient treatment process. The automated staging, completed by a computer program, takes a mere 5 s (C++, NVIDIA GeForce RTX 3070), while manual staging requires at least 30 min of a technician’s time. The superiority of the automated approach is further validated by evaluation from experienced clinicians who confirmed its superior performance compared to manual approaches. Additionally, the algorithm’s strict adherence to biomechanical principles through force balance calculations provides a level of mathematical precision that would be impractical to achieve through manual staging.

## 4. Discussion

The clear aligner treatment can be simplified by applying equal-magnitude, oppositely directed forces and moments to each tooth, effectively guiding them into the desired alignment. This approach not only demonstrates the aligners’ capability to correct tooth positions but also adheres to the fundamental principles of physics, ensuring that the net sum of all forces and moments exerted by the aligners on the dentition is zero, as represented by the following equations:(14)$$F_{1}+F_{2}+\cdots +F_{14}=0$$(15)$$M_{1}+M_{2}+\cdots +M_{14}=0$$

The proposed method offers a more straightforward solution compared to artificial intelligence or path planning algorithms. By directly utilizing the forces acting on the teeth, this force-driven approach adaptively and flexibly resolves tooth collision issues. In contrast, other methods necessitate setting various rules to handle collisions, resulting in increased complexity and difficulty in finding optimal solutions.

Aligner orthodontic treatment primarily focuses on aligning the relative positions of teeth rather than their absolute positions in space. In common malocclusions, such as Class II and dental arch deviation (Figure 8), the proposed method cannot directly align the teeth to the target position if the relative positions of the initial and target teeth are identical. To overcome this limitation, an anchorage control and sequential movement approach is necessary. For instance, in distal movement, most teeth should serve as anchorage while a few teeth move during staging. The most common pattern is the distalization 1/2 pattern (Figure 9a), where each tooth moves halfway before engaging the next. However, due to the first molar’s large root surface area, which may lead to some degree of anchorage loss, a more conservative approach involves moving it independently during distal movement (Figure 9b). For cases requiring entire arch displacement, sequential movements with anchorage control should be superimposed. To ensure force equilibrium and anchorage control on teeth (Figure 10a), corresponding overcorrection design should be added to the aligners (Figure 10b). Figure 11 compares the force-driven automated staging with doctor-guided manual staging for a molar distalization case. The number of stages for force-driven automated staging is 30 for the maxilla and 34 for the mandible [31], while for doctor-guided manual staging, it is 41 for both the maxilla and mandible [32]. In comparison, automated staging demonstrates significant advantages.

In the previously discussed model, the forces and moments were considered to cause independent translations and rotations. However, when an aligner applies a force on the dental crown, it causes the tooth to tip (Figure 12a). To prevent this and maintain tooth translation, an opposing moment (Figure 12b) must be applied by designing the aligner with an overcorrection of reverse rotation (Figure 12c). The magnitude of this overcorrection depends on the distance between the applied force and the tooth’s center of resistance [33] and can be determined through finite element analysis and experimental validation.

Despite its effectiveness in simplifying clear aligner treatment planning, the force-driven model’s accuracy in calculating tooth translation and rotation depends on multiple critical factors: aligner material properties, periodontal ligament (PDL) characteristics, tooth morphology, attachment configurations, jaw relationships, patient age, and interproximal reduction (IPR) timing. Integration of these factors into the force-driven model enables continuous refinement and optimization of stiffness coefficients. However, the current model faces challenges in addressing comprehensive dental arch discrepancies, particularly midline deviations and transverse discrepancies, which often necessitate sophisticated treatment strategies including overcorrection and auxiliary anchorage devices. Future advancement of this approach requires artificial intelligence integration to automatically optimize parameters for individual cases and incorporate diverse treatment modalities [34,35,36], ultimately enhancing the model’s versatility and clinical effectiveness in clear aligner therapy.

Beyond these model-specific challenges, the broader evolution of clear aligner technology faces limitations not from manufacturing capabilities, which have significantly advanced through direct 3D printing of aligners and in-house production, but rather from the complexity of staging design. The implementation of force-driven automated staging methodology represents a transformative solution that extends beyond clinical benefits. This approach can dramatically streamline the workflow of clear aligner manufacturers by reducing reliance on technical staff for manual staging design, thereby optimizing production efficiency, shortening manufacturing cycles, and lowering operational costs. This technological advancement could reshape the competitive landscape of the clear aligner industry, challenging the market dominance of major manufacturers and potentially leading to reduced market concentration and more competitive pricing structures. Moreover, the optimization of staging sequences through force-driven automation typically results in fewer required stages while maintaining treatment efficacy, translating to reduced treatment costs for patients. This democratization of clear aligner therapy through improved accessibility and affordability could have far-reaching implications for public health and social welfare, making advanced orthodontic care available to a broader demographic.

## 5. Conclusions

The force-driven automated staging design introduced in this study represents an initial step towards automated clear aligner treatment planning, demonstrating potential improvements in efficiency compared to traditional manual staging methods. By integrating fundamental principles of aligner biomechanics and physical laws, this automated staging method aims to ensure the feasibility and rationality of each generated stage. In cases requiring entire arch displacement, maintaining force equilibrium while superimposing sequential movements with anchorage control is essential. While our current approach focuses on specific biomechanical principles, we strongly emphasize that the inherent complexity of orthodontic treatment necessitates the development of diverse techniques to address the broad spectrum of clinical scenarios. Further biomechanical analyses and experimental validations are needed to refine the model’s parameters and develop additional methodologies, enabling comprehensive solutions for diverse orthodontic cases and enhancing the accuracy of automated staging designs.

## Figures and Tables

**Figure 1 bioengineering-12-00111-f001:**
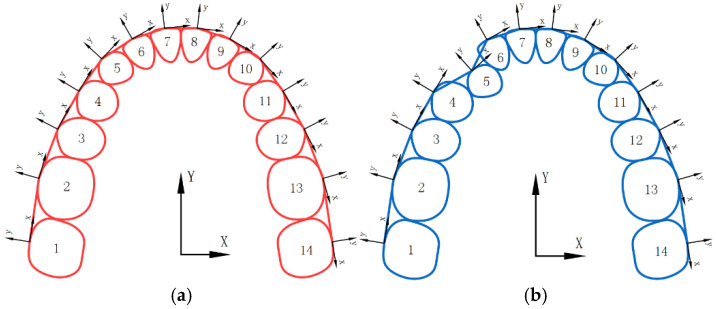
(**a**) The target position; (**b**) the current position.

**Figure 2 bioengineering-12-00111-f002:**
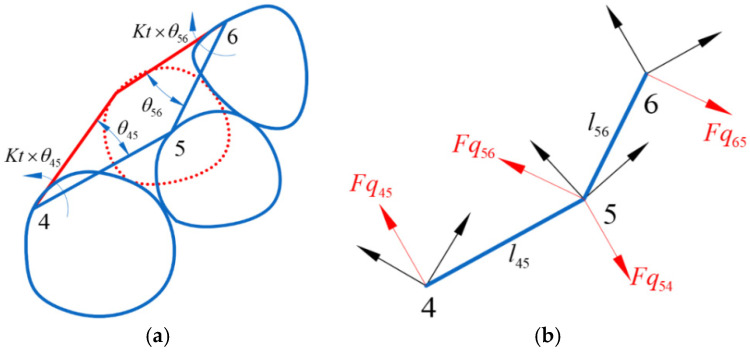
(**a**) Angle between the lines connecting the current and target tooth positions; (**b**) forces generated by the angle correction; (**c**) the tensile force between teeth; (**d**) the collision force between teeth; (**e**) the resultant force on teeth 5.

**Figure 3 bioengineering-12-00111-f003:**
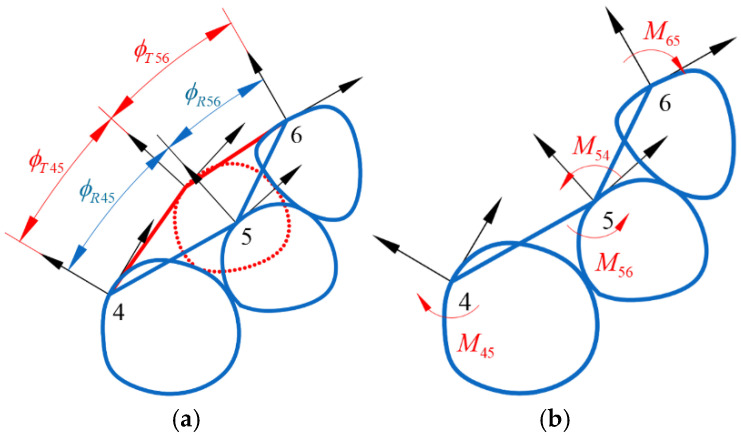
(**a**) Angle between adjacent teeth in the current position versus the target position; (**b**) moments generated by angular discrepancy.

**Figure 4 bioengineering-12-00111-f004:**
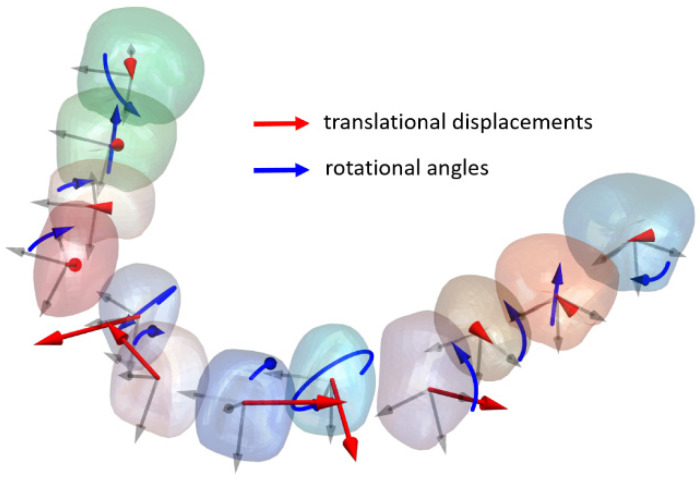
Stepwise translational displacements and rotational angles of teeth in 3D space.

**Figure 5 bioengineering-12-00111-f005:**
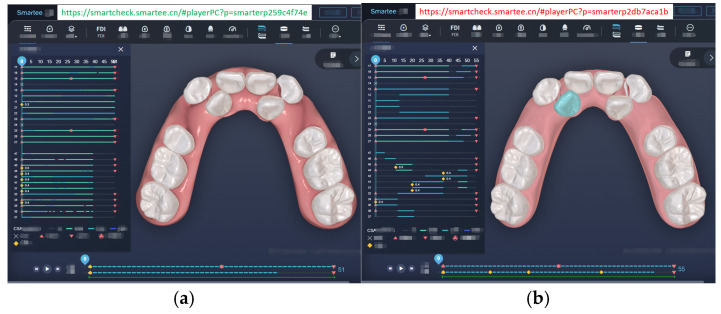
(**a**) Force-driven automated staging (https://smartcheck.smartee.cn/#playerPC?p=smarterp259c4f74e (accessed on 10 September 2024)); (**b**) doctor-guided manual staging (https://smartcheck.smartee.cn/#playerPC?p=smarterp2db7aca1b (accessed on 10 September 2024)).

**Figure 6 bioengineering-12-00111-f006:**
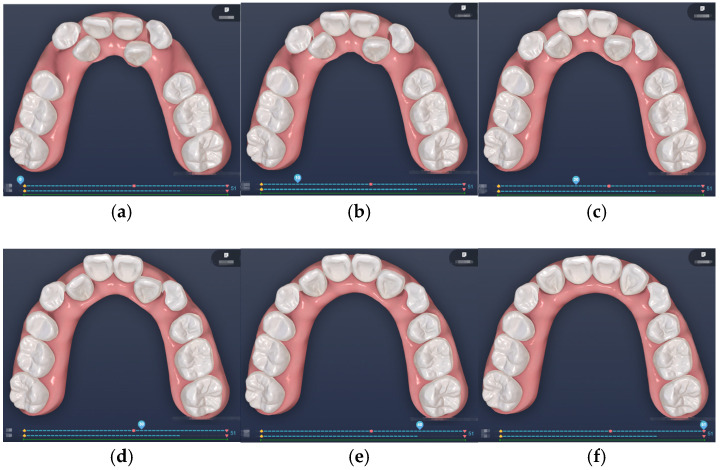
Force-driven automated staging: (**a**) Stage 0; (**b**) Stage 10; (**c**) Stage 20; (**d**) Stage 30; (**e**) Stage 40; (**f**) Stage 51.

**Figure 7 bioengineering-12-00111-f007:**
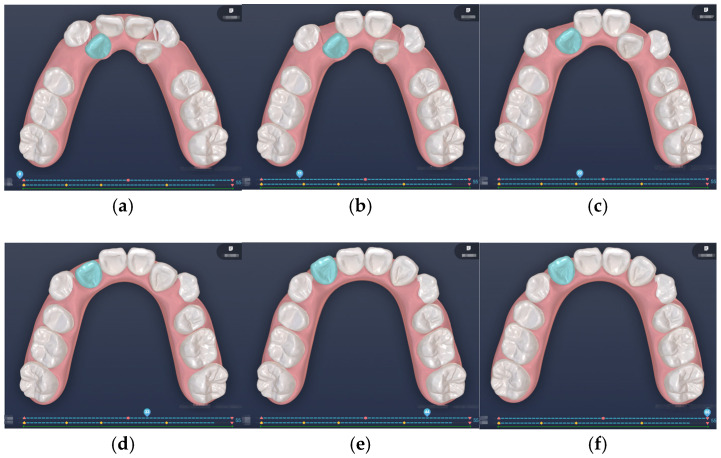
Doctor-guided manual staging: (**a**) Stage 0; (**b**) Stage 11; (**c**) Stage 22; (**d**) Stage 33; (**e**) Stage 44; (**f**) Stage 55.

**Figure 8 bioengineering-12-00111-f008:**
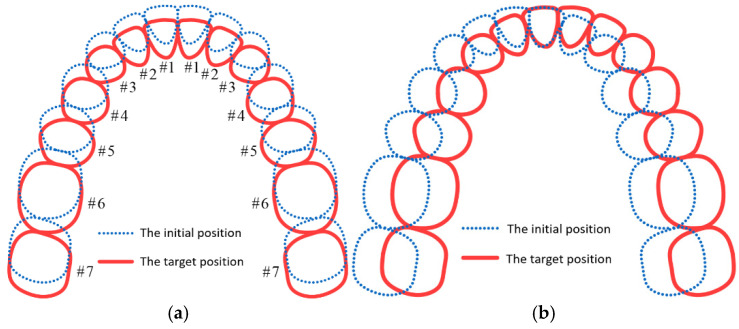
(**a**) Class II malocclusion; (**b**) dental arch deviation.

**Figure 9 bioengineering-12-00111-f009:**
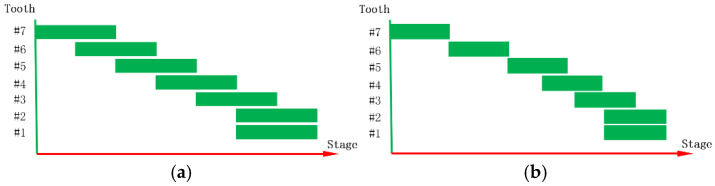
(**a**) Distalization 1/2 pattern; (**b**) conservative distalization pattern.

**Figure 10 bioengineering-12-00111-f010:**
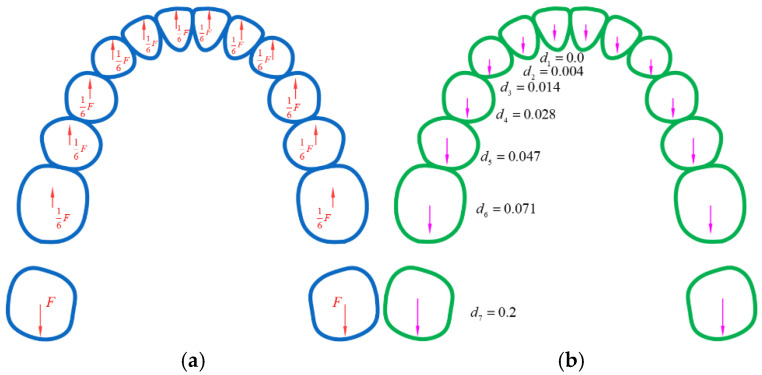
(**a**) Force equilibrium of teeth during molar distalization; (**b**) corresponding overcorrection design of aligner displacement.

**Figure 11 bioengineering-12-00111-f011:**
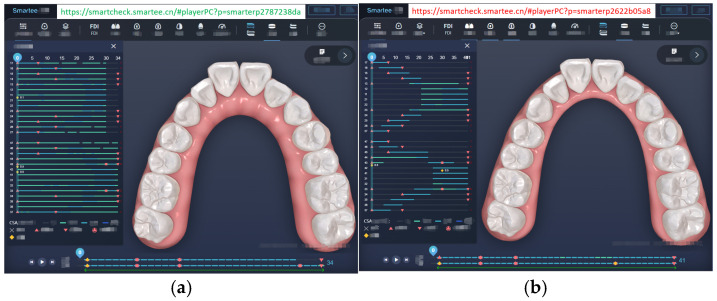
(**a**) Force-driven automated staging for molar distalization (https://smartcheck.smartee.cn/#playerPC?p=smarterp2787238da (accessed on 10 September 2024)); (**b**) doctor-guided manual staging for molar distalization (https://smartcheck.smartee.cn/#playerPC?p=smarterp2622b05a8 (accessed on 10 September 2024)).

**Figure 12 bioengineering-12-00111-f012:**
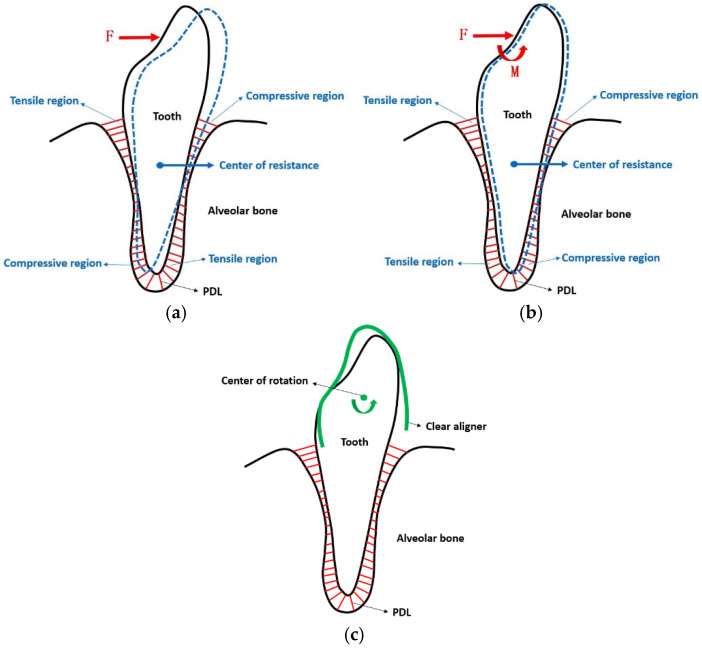
(**a**) Force acting on the dental crown; (**b**) force and moment acting on the dental crown; (**c**) corresponding overcorrection rotation of aligner.

## Data Availability

Data are contained within the article and Appendix A.

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
