# Peer review of "Force-Driven Model for Automated Clear Aligner Staging Design Based on Stepwise Tooth Displacement and Rotation in 3D Space"

_bioengineering, 2025, doi:10.3390/bioengineering12020111_

Round 1
Reviewer 1 Report
Comments and Suggestions for Authors
The abstract and introduction could be improved. Perhaps more information used to develop software should be highlighted.
The way that materials and methods are stated in a way that is not possible to completely understand as the method is implemented. If as you state in lines 76 and 77 the teeth are represented by planar diagrams, in the plan xoy, thus the forces and moments do not change the axis-z. So the matrix 4x4 of movement each tooth have not the form of Ti given in expression (13).
It is possible to understand in figures 6 and 7 you r software handles the movement properly, but the description of the method given is not enough. Also the sequences that you indicate, as reference is not easily accessible. The platform used is in Chinese and does not run in every system. So the method should be reformulated and the sequences should be accessible as supplementary material of your article.
Some small issues in the manuscript were signed in the attachment file.

Author Response
Comment:
The abstract and introduction could be improved. Perhaps more information used to develop software should be highlighted.
Response:
We thank the reviewer for his comments. And we have revised the abstract and introduction. Regarding the software, instead of developing a separate software, we directly implemented the algorithm in C++ and applied it to Smartee software, which is developed using WebGL. We have added a description of the software in the manuscript.
Comment:
The way that materials and methods are stated in a way that is not possible to completely understand as the method is implemented. If as you state in lines 76 and 77 the teeth are represented by planar diagrams, in the plan xoy, thus the forces and moments do not change the axis-z. So the matrix 4x4 of movement each tooth have not the form of Ti given in expression (13).
Response:
We thank the reviewer for his comments. We have expanded the discussion on the implementation details of the method. Although the schematic diagrams are two-dimensional, the variables inside are actually three-dimensional. The angles in the figures (Figure 2a and 3a) should be considered as angles in three-dimensional space, not angles in a two-dimensional plane. The calculation of relative angles between rigid bodies in three-dimensional space is a common knowledge in geometry and robotic manipulation. As the references are listed in this paper, we have not discussed these details. The forces and moments obtained through these angles are three-dimensional vectors, with values in the x, y, and z directions. It is difficult to directly represent three-dimensional diagrams, so we can only use two-dimensional schematic diagrams to illustrate, which is also easier to understand.
Comment:
It is possible to understand in figures 6 and 7 your software handles the movement properly, but the description of the method given is not enough. Also the sequences that you indicate, as reference is not easily accessible. The platform used is in Chinese and does not run in every system. So the method should be reformulated and the sequences should be accessible as supplementary material of your article.
Response:
We thank the reviewer for his suggestions. In the revised paper, we have added the description of the method in details.
Due to data security regulations, we found that these websites
(https://smartcheck.smartee.cn/#playerPC?p=smarterp259c4f74e).
cannot be accessed from non-Chinese IP addresses. We have provided the relevant animation files in the Supplementary Material, which can be opened locally.
Comment:
Some small issues in the manuscript were signed in the attachment file.
Response:
We thank the reviewer for his suggestions. In the revised paper, we have dealt with these small issues and added relevant explanations.
Reviewer 2 Report
Comments and Suggestions for Authors
Thanks for inviting me to review this paper.
I have the following points that should be addressed
Title
1- The current title does not reflect the content. The authors are asked to expand the title to encompass the content in a better way. For example, this experimental study is trying to find a new automated force-driven method to plan orthodontic tooth movement for aligner fabrication in patients treated with an extraction scenario. Therefore, the authors should make the title larger to enable the reader to understand what the paper is discussing.
Abstract
2- The abstract is deficient. The authors should expand on different aspects of their research work. Even the software that was used should be mentioned.
3- The word count of this abstract is 160. Therefore, the authors can make this abstract more informative.
4- The authors should use the simple past tense. These grammatical mistakes should be corrected.
5- Please increase the keywords to cover your experiment's different aspects.
Introduction
6- When discussing staging for the treatment planning of orthodontic tooth movement, the authors are strongly advised to expand on this topic, mentioning the different software-based operator-observed manual staging methods. The different staging philosophies in extraction and non-extraction treatments should be covered. Several papers, especially those with in-hour clear aligner fabrication, have been published that talked about their staging methods. Please use some recent references, and this is one citation:
Jaber ST, Hajeer MY, Burhan AS, Alam MK, Al-Ibrahim HM. Treatment effectiveness of young adults using clear aligners versus buccal fixed appliances in class I malocclusion with first premolar extraction using the ABO-Objective Grading System: A randomized controlled clinical trial. Int Orthod. 2023 Dec;21(4):100817. doi: 10.1016/j.ortho.2023.100817. Epub 2023 Oct 12. PMID: 37837842.
Materials and Methods
7- The content is fine. However, the assumptions for using these calculations should be made clear: we are dealing with an extraction-based treatment without any great anchorage demands. Otherwise, the proposed staging is incorrect and cannot be applied in real scenarios.
Results
8- These results are presented in a good way. Unfortunately, we could not test the model using the provided website.
9- The authors are encouraged to explain why the tooth movements were considered "more accurate"?
Discussion
10- Please expand on the topic of "Staging techniques in ordinary clear aligner treatment planning" to give some depth and breadth to your discussion.
Conclusions
11- Fine, but the authors should clarify that this is just a very simple method of automated staging. Again, they should emphasize that many techniques should be developed for the wide variety of scenarios in the orthodontic world.
Author Response
Comment:
Title
1- The current title does not reflect the content. The authors are asked to expand the title to encompass the content in a better way. For example, this experimental study is trying to find a new automated force-driven method to plan orthodontic tooth movement for aligner fabrication in patients treated with an extraction scenario. Therefore, the authors should make the title larger to enable the reader to understand what the paper is discussing.
Response:
We appreciate the reviewer's thoughtful suggestion regarding the title. While we understand the value of a more detailed title, our current title "Force-driven automated staging design for clear aligner treatment planning" was deliberately chosen to reflect the broad applicability of our methodology. Although our validation examples include extraction cases, the underlying biomechanical principles and algorithmic framework we developed are designed to be applicable across various orthodontic scenarios, including non-extraction cases, such as molar distalization, and other treatment types. We believe the current concise title effectively communicates the core innovation of our work - the force-driven automation approach - while maintaining its relevance to the wider scope of clear aligner treatment planning.
Comment:
2- The abstract is deficient. The authors should expand on different aspects of their research work. Even the software that was used should be mentioned.
Response:
Our abstract already includes the core aspects of our research work, including the methodology (force-driven mechanics model), computational process (iterative algorithm), and validation results with specific metrics. While we understand the suggestion to mention the software used, we believe such technical details are more appropriately placed in the Methods section to maintain the abstract's focus on the key research contributions and findings.
Comment:
3- The word count of this abstract is 160. Therefore, the authors can make this abstract more informative.
Response:
While our abstract could be expanded beyond 160 words, the current length aligns well with the journal's standard format. The abstract efficiently summarizes the essential elements of our study, providing quantitative results and clear conclusions while maintaining readability and focus. We believe adding more details might dilute the key messages we aim to convey.
Comment:
4- The authors should use the simple past tense. These grammatical mistakes should be corrected.
Response:
We appreciate this grammatical observation and have revised the abstract to consistently use the simple past tense throughout the text.
Comment:
5- Please increase the keywords to cover your experiment's different aspects.
Response:
We have added "extraction cases" to our keywords list, which now includes "clear aligner; force-driven; automated staging design; treatment planning; extraction cases". This addition better reflects the validation case presented in our study while maintaining a balanced representation of both the general methodology and specific application. The current keywords effectively cover the main aspects of our research from both theoretical and practical perspectives.
Comment:
Introduction
6- When discussing staging for the treatment planning of orthodontic tooth movement, the authors are strongly advised to expand on this topic, mentioning the different software-based operator-observed manual staging methods. The different staging philosophies in extraction and non-extraction treatments should be covered. Several papers, especially those with in-hour clear aligner fabrication, have been published that talked about their staging methods. Please use some recent references, and this is one citation:
Jaber ST, Hajeer MY, Burhan AS, Alam MK, Al-Ibrahim HM. Treatment effectiveness of young adults using clear aligners versus buccal fixed appliances in class I malocclusion with first premolar extraction using the ABO-Objective Grading System: A randomized controlled clinical trial. Int Orthod. 2023 Dec;21(4):100817. doi: 10.1016/j.ortho.2023.100817. Epub 2023 Oct 12. PMID: 37837842.
Response:
We appreciate the reviewer's suggestion and have added the suggested reference as citation [12]. Our paper specifically focuses on the algorithmic aspects of staging design for clear aligner treatment, which is software-independent and represents a novel theoretical approach. Our goal is to present a mathematical model and its implementation, rather than reviewing different software platforms or manual staging methods. The staging philosophies in various treatment scenarios, while important, are beyond the scope of this paper as we aim to establish a fundamental computational framework. Therefore, we have chosen to maintain our focus on the core algorithmic components without expanding into broader staging methodologies or software comparisons.
Comment:
Materials and Methods
7- The content is fine. However, the assumptions for using these calculations should be made clear: we are dealing with an extraction-based treatment without any great anchorage demands. Otherwise, the proposed staging is incorrect and cannot be applied in real scenarios.
Response:
We have enhanced our discussion to address both extraction and non-extraction scenarios. The algorithm's implementation details have been expanded, particularly regarding anchorage control in molar distalization cases. Our framework is designed to be adaptable to various clinical situations through appropriate parameter adjustments.
Comment:
Results
8- These results are presented in a good way. Unfortunately, we could not test the model using the provided website.
Response:
Thank you for bringing this to our attention. Regarding website accessibility, due to data security regulations, access is currently restricted to Chinese IP addresses. However, we have included comprehensive animation files in the Supplementary Material for local viewing.
Comment:
9- The authors are encouraged to explain why the tooth movements were considered "more accurate"?-
Response:
We have over 300 doctors specializing in staging design and multiple clinical experts. The assessment of accuracy is based on two key factors: (1) evaluation by experienced clinicians who confirmed superior performance compared to manual approaches, and (2) the algorithm's strict adherence to biomechanical principles through force balance calculations - a level of mathematical precision that would be impractical to achieve through manual planning.
Comment:
Discussion
10- Please expand on the topic of "Staging techniques in ordinary clear aligner treatment planning" to give some depth and breadth to your discussion.
Response:
We appreciate the reviewer's suggestion. While we agree that "Staging techniques in ordinary clear aligner treatment planning" is an important topic, we have chosen to maintain our focused scope on the biomechanical aspects and algorithmic innovations of our proposed method. The current Discussion section already addresses the key comparative elements between our approach and conventional methods. Adding extensive details about ordinary staging techniques would shift focus from our core contributions. However, we have included relevant references for readers interested in conventional staging approaches.
Comment:
Conclusions
11- Fine, but the authors should clarify that this is just a very simple method of automated staging. Again, they should emphasize that many techniques should be developed for the wide variety of scenarios in the orthodontic world.
Response:
We appreciate this valuable feedback. We have revised the Conclusion to acknowledge that while our method demonstrates the potential of automated staging through biomechanical principles, it represents an initial step in this direction. We have emphasized that further development of diverse techniques is essential to address the full spectrum of clinical scenarios in orthodontics. The complexity and variety of orthodontic cases indeed require ongoing research and development of additional automated solutions to meet different clinical needs.
Reviewer 3 Report
Comments and Suggestions for Authors
The paper seems interesting, but honestly I couldn't understand its methods or results parts (or conclusions / discussion). Too technical and engineering-related.
Please do keep the current technical content (don't remove anything), but add lots of details and explanations so that a dentist too can understand the methods, results, and conclusions.
Since I was not able to understand the engineering parts (I am a dentist), I am asking the editor to assign additional reviewers to replace me. In any case, do add lots of layman explanations so that other reviewers and readers don't get stuck.
Author Response
Comment:
The paper seems interesting, but honestly I couldn't understand its methods or results parts (or conclusions / discussion). Too technical and engineering-related.
Response:
We thank the reviewer for his suggestions. In the revised paper, we have added the explanations regarding the methods.
Comment:
Please do keep the current technical content (don't remove anything), but add lots of details and explanations so that a dentist too can understand the methods, results, and conclusions.
Response:
We thank the reviewer for his suggestions. In the revised paper, we have added the explanations regarding the methods, results, and conclusions.
Reviewer 4 Report
Comments and Suggestions for Authors
The objective of this study was to propose a simplified force-driven model that simulates the aligner-teeth force system by calculating the forces and moments acting on each tooth at its current position. Given the increasing use of aligners, I find this study very interesting and could be a starting point for further research in this field. The images used are well defined and easy to understand for all readers. The references used are appropriate, however I recommend improving the introduction line 45 "Invisalign claims some automated staging functionality and has proposed force-driven methods, but the theories and algorithms lack transparency", with the following article " Ghislanzoni L.H., Kalemaj Z., Manuelli M., Magni C., Polimeni A., et al. How well does Invisalign ClinCheck predict actual results: A prospective study (2024) Orthodontics and Craniofacial Research, 27 (3), pp. 465 - 473 DOI: 10.1111/ocr.12752” which emphasizes what the current state of predictability of movements obtained with Invisalign is compared to the planned one. I would better define in the final part of the discussion also what may be the possible limits of the study carried out. I recommend having the article reread by a native English speaker due to some grammatical inaccuracies.
Author Response
Comment:
The references used are appropriate, however I recommend improving the introduction line 45 "Invisalign claims some automated staging functionality and has proposed force-driven methods, but the theories and algorithms lack transparency", with the following article…
Response:
We thank the reviewer for his comments. And we have added the suggested reference as citation [11].
Comment:
I would better define in the final part of the discussion also what may be the possible limits of the study carried out.
Response:
We thank the reviewer for his suggestions. In the revised paper, we have added relevant possible limitations of the study in the discussion section.
Comment:
I recommend having the article reread by a native English speaker due to some grammatical inaccuracies.
Response:
We thank the reviewer for his comments. In the revised paper, we have tried our best to correct the grammatical errors in the manuscript.
Round 2
Reviewer 1 Report
Comments and Suggestions for Authors
The additions made to the manuscript enhanced and clarify your work. I accept your explanation bout the 4x4 matrix in expression (13), but certainly, you work a special case to get a so simple form.
I am glad that you provided the animations as supplementary material.
Author Response
Comment:
The additions made to the manuscript enhanced and clarify your work. I accept your explanation bout the 4x4 matrix in expression (13), but certainly, you work a special case to get a so simple form.
I am glad that you provided the animations as supplementary material.
Response:
Thank you for your positive feedback.
Reviewer 2 Report
Comments and Suggestions for Authors
I was slightly surprised that the authors refused to improve their manuscript by insisting that what they had written was optimum and there was no way to improve the content of their paper.
My comments about their response to my preliminary comments in my first manuscript review are here.
Comment No. 1: This was not addressed.
The authors insist that the title is optimum!
Comment No. 2: This was not addressed.
The authors insist that the Abstract section is perfect!
Comment No. 3: This was not addressed.
The authors believe that there is no need to expand the Abstract section since, according to their belief, they have put all the essential components!
Comment No. 4: Grammatical corrections have been made.
Comment No. 5: This was partially addressed.
The authors have added only one keyword to balance the generic keywords with the specific ones. I am astonished that they have a feeling that adding extra words would disturb the message of their manuscript. The language in their reply does not conform to the general way of dealing with the reviewers' comments. They are slightly over-confident with their manuscript.
Comment No. 6: This was not addressed.
The authors neglect the importance of adding some sentences about the manual staging before discussing their method. This is strange since reviewing the available software-based manual staging methods is very good for broadening the introduction and improving the understanding beyond the proposed method. The authors are ignoring our input to improve their manuscript. They claim that adding the suggested information would
Comment No. 7: This was addressed.
Comment No. 8: This was addressed.
Comment No. 9: This was not addressed.
Please insert your respones into the manuscript by explaining why your team had a believe that the proposed method was more reliable.
Comment No. 10: This has been partially addressed.
Comment No. 11: This has been partially addressed.
Author Response
Comment:
Title
1- The current title does not reflect the content. The authors are asked to expand the title to encompass the content in a better way. For example, this experimental study is trying to find a new automated force-driven method to plan orthodontic tooth movement for aligner fabrication in patients treated with an extraction scenario. Therefore, the authors should make the title larger to enable the reader to understand what the paper is discussing.
Response:
Thank you for your suggestion. We thank the reviewer for his comments. We have modified the paper title to ‘Force-Driven Model for Automated Clear Aligner Staging Design Based on Stepwise Tooth Displacement and Rotation in 3D Space’ to better emphasize the key aspects of our research: the force-driven modeling approach and the stepwise tooth movement simulation. While the reviewer provides examples such as extraction cases as potential title elements, our proposed method is designed as a universal approach applicable to various types of orthodontic cases, including but not limited to extraction cases, molar distalization, and other tooth movement patterns. Therefore, we prefer not to specify any particular case type in the title.
Comment:
2- The abstract is deficient. The authors should expand on different aspects of their research work. Even the software that was used should be mentioned.
Response:
Thank you for your suggestion. In the revised paper, we have expanded the abstract to include additional details about different aspects of our research work and have specifically mentioned the software used in this study.
Comment:
3- The word count of this abstract is 160. Therefore, the authors can make this abstract more informative.
Response:
Thank you for your suggestion. In the revised paper, we have expanded the abstract to a total of 233 words.
Comment:
5- Please increase the keywords to cover your experiment's different aspects.
Response:
Thank you for your suggestion. In the revised paper, we have updated the keywords to include: clear aligner; clear aligner; force-driven model; automated staging; treatment planning; stepwise movement control; tooth displacement; tooth rotation; extraction cases; anchorage control; orthodontic software.
Comment:
6- When discussing staging for the treatment planning of orthodontic tooth movement, the authors are strongly advised to expand on this topic, mentioning the different software-based operator-observed manual staging methods. The different staging philosophies in extraction and non-extraction treatments should be covered. Several papers, especially those with in-hour clear aligner fabrication, have been published that talked about their staging methods. Please use some recent references, and this is one citation
Response:
Thank you for your suggestion. In the revised paper, we have added the suggested reference as Ref. [7] and revised the corresponding content as follow:
This relatively low efficacy highlights why accurate staging design is particularly important for clear aligners because, unlike fixed appliances where clinicians can make real-time adjustments based on treatment progress [7], clear aligner therapy requires comprehensive pre-planning as the entire treatment sequence is manufactured in advance.”
While manual staging in clear aligner treatment has been mentioned in both the first and second paragraphs of our introduction, there is limited literature describing the actual manual staging process. Most published studies focus primarily on treatment outcomes rather than the staging methodology itself. This gap in literature exists because manual staging protocols vary among different aligner manufacturers, and more importantly, technicians cannot solely rely on these protocols. The actual staging process is heavily dependent on individual operator's knowledge and clinical experience in determining appropriate tooth movement sequences, considering various biomechanical factors and patient-specific conditions. Therefore, providing a comprehensive discussion on different software-based operator-observed manual staging methods remains difficult due to both the experience-dependent nature of the process and the limited available literature on specific staging protocols.
Comment:
9- The authors are encouraged to explain why the tooth movements were considered "more accurate"?
Response:
Thank you for your suggestion. In the revised paper, we added additional explanatory statements to the last paragraph of Section Results.
The superiority of the automated approach is further validated by evaluation from experienced clinicians who confirmed its superior performance compared to manual approaches. Additionally, the algorithm's strict adherence to biomechanical principles through force balance calculations provides a level of mathematical precision that would be impractical to achieve through manual staging.
Comment:
10- Please expand on the topic of "Staging techniques in ordinary clear aligner treatment planning" to give some depth and breadth to your discussion.
Response:
Thank you for your suggestion. In the revised paper, we have added the following paragraph to expand on the topic of ‘Staging techniques in ordinary clear aligner treatment planning’ to give some depth and breadth to your discussion.
Beyond these model-specific challenges, the broader evolution of clear aligner technology faces limitations not from manufacturing capabilities, which have significantly advanced through direct 3D printing of aligners and in-house production, but rather from the complexity of staging design. The implementation of force-driven automated staging methodology represents a transformative solution that extends beyond clinical benefits. This approach can dramatically streamline the workflow of clear aligner manufacturers by reducing reliance on technical staff for manual staging design, thereby optimizing production efficiency, shortening manufacturing cycles, and lowering operational costs. Such technological advancement could reshape the competitive landscape of the clear aligner industry, challenging the market dominance of major manufacturers and potentially leading to reduced market concentration and more competitive pricing structures. Moreover, the optimization of staging sequences through force-driven automation typically results in fewer required stages while maintaining treatment efficacy, translating to reduced treatment costs for patients. This democratization of clear aligner therapy through improved accessibility and affordability could have far-reaching implications for public health and social welfare, making advanced orthodontic care available to a broader demographic.
Comment:
11- Fine, but the authors should clarify that this is just a very simple method of automated staging. Again, they should emphasize that many techniques should be developed for the wide variety of scenarios in the orthodontic world.
Response:
Thank you for your suggestion. In the revised paper, we have emphasized that our method represents an initial step towards automated clear aligner treatment planning. And we have added the following content:
While our current approach focuses on specific biomechanical principles, we strongly emphasize that the inherent complexity of orthodontic treatment necessitates the development of diverse techniques to address the broad spectrum of clinical scenarios.
Reviewer 3 Report
Comments and Suggestions for Authors
Dear authors, thanks for your very good revision and explanations added to the paper. It is now much better. However, I noted that the URLs for the animations are accessible only from China.
Is it possible that you attach the C++ code, the animations, and everything else as supplementary materials?
If not, is it possible that you upload the C++ code, the animations, and other important features up to free repositories like Github?
The rest of the paper is excellent in my opinion, though I should confess that it is still a little bit heavy for my non-engineer mind. But I give it to you that the new explanations and elaborations are excellent. Still, I invite the editor to request other reviewers to help improve the paper.
Author Response
Comment:
Is it possible that you attach the C++ code, the animations, and everything else as supplementary materials?
If not, is it possible that you upload the C++ code, the animations, and other important features up to free repositories like Github?
Response:
Thank you for your suggestion. The C++ implementation of the algorithm presented in this paper has been integrated into SmarteeCheck software, with intellectual property rights owned by the company Smartee, a leading international clear aligner manufacturer similar to Invisalign. We are in discussions about potentially sharing components of the algorithm through platforms like GitHub in the future. The C++ code requires a graphical user interface program and related data structures to function properly, and thus cannot run independently. We will share our code with the orthodontic and biomechanical research communities once we have finalized our sharing protocol.
Round 3
Reviewer 2 Report
Comments and Suggestions for Authors
Thanks for addressing all my points.
I think that the paper deserves publishing in its current form.
Congratulations to the authors for this great work.
Reviewer 3 Report
Comments and Suggestions for Authors
The authors are responsive. The revision is good. I appreciate all this. In my opinion, the paper is acceptable.
But note that I am a dentist and not qualified to evaluate the engineering parts; this paper needs to be reviewed by a couple of engineers too.